# Waldenström Macroglobulinemia: Mechanisms of Disease Progression and Current Therapies

**DOI:** 10.3390/ijms231911145

**Published:** 2022-09-22

**Authors:** Ava J. Boutilier, Lina Huang, Sherine F. Elsawa

**Affiliations:** 1Department of Molecular, Cellular and Biomedical Sciences, University of New Hampshire, Durham, NH 03824, USA; 2Phillips Exeter Academy, Exeter, NH 03833, USA

**Keywords:** Waldenström macroglobulinemia, B-cell lymphoma, non-Hodgkin’s, bone marrow microenvironment, disease progression, targeted therapy

## Abstract

Waldenström macroglobulinemia is an indolent, B-cell lymphoma without a known cure. The bone marrow microenvironment and cytokines both play key roles in Waldenström macroglobulinemia (WM) tumor progression. Only one FDA-approved drug exists for the treatment of WM, Ibrutinib, but treatment plans involve a variety of drugs and inhibitors. This review explores avenues of tumor progression and targeted drug therapy that have been investigated in WM and related B-cell lymphomas.

## 1. Introduction

Waldenström macroglobulinemia (WM) is a rare, indolent B-cell malignancy, characterized by the infiltration of plasma cells, plasmacytoid lymphocytes, and small lymphocytes to the bone marrow [1,2,3]. The median age of diagnosis is 63–68 years old [4] and this disease accounts for 1–2% of hematological neoplasms, with an age-adjusted incidence rate of 3.4 per million among male and 1.7 per million among female populations in the United States [4]. The World Health Organization defines WM as a lymphoplasmacytic lymphoma (LPL) with an immunoglobulin M (IgM) paraprotein [2,5]. The most frequently observed cytogenetic abnormality in WM is deletion of the long arm of chromosome 6 (6q). This deletion is correlated with poor prognostic features, such as higher levels of beta2-microglobulin and a greater prevalence of hypoalbuminemia and anemia [4,6]. There is no standard treatment for WM and treatment programs are variable. To date, no cure has been discovered for WM [7], although allogeneic hematopoietic stem cell transplantation (allo-HSCT) has become a cornerstone in the treatment of hematological malignancies such as WM. Within allo-HSCT therapy, there is still a problem with disease relapse, and it is associated with poor long-term survival [8]. Because most WM/LPL patients eventually relapse, there is a need to balance the benefits of treatment and the side effects of that treatment. Some patients with WM will relapse with the aggressive form of non-Hodgkin lymphoma [9,10]. Approximately 75% of WM patients have symptoms at the time of diagnosis and anemia is common, as WM cell growth in the bone marrow progresses [2]. Hyperviscosity syndrome, caused by abnormal IgM secretion and accumulation in the blood, occurs in 10–30% of patients and may be life-threatening [2].

Discriminating WM from chronic lymphocytic leukemia (CLL), mantle cell lymphoma (MCL), and follicular lymphoma involves noting the presence of CD19, CD20, CD22, CD79α, and CD138 cell expression and lack of CD5, CD10, and CD23. The expression of CD5, CD10, and CD23 may be found in 10–20% of lymphoma cases and does not exclude the diagnosis of WM. WM has to be separated from the CD5+ lymphoplasmocytoid lymphoma as these cases are B-CLL variants. A diagnosis of WM requires the differentiation from IgM myeloma, which is a rare disease presenting as a homogenous plasma cell population in the bone marrow, characterized by symptomatic clonal plasma cell proliferation, 10% or more plasma cells on bone marrow biopsy, plus the presence of lytic bone lesions and/or IGH-translocations 4p16, 6p21, 11q13, 16q23, and 20q11 [11].

WM has an associated precursor disease, monoclonal gammopathy of undetermined significance (MGUS) of IgM type. IgM MGUS is defined by a serum concentration of 3 g/dL or less of IgM paraprotein, the absence of proliferation of plasma cells, and a population of 10% or less of bone marrow plasma cells. MGUS can progress to WM, and typically does so at a rate of 1% per year, indicating the importance of understanding IgM MGUS. Several prognostic markers indicating the risk of disease progression have been proposed. Due to the relatedness of these two diseases, it can be clinically difficult to distinguish these two diseases. One proposed marker has shown some success in distinguishing between MGUS and WM, the presence of a deletion of the long arm of chromosome 6 (6q), as no patients IgM MGUS patients with a 6q-deletion have been documented [11].

## 2. Genetic Abnormalities

Deletion of the 6q chromosome appears to be the most common abnormality in WM, occurring in up to 50% of patients [12]. A study in 2006 showed that patients with the deletion of 6q required treatment more frequently and displayed a shorter treatment-free survival, compared with patients without the deletion [6]. The survival analysis, however, showed that there was no significant difference between the 6q deletion group and the non-deletion group in terms of median survival [6]. Another study, conducted in 2013, investigated different chromosomal aberrations. The main aberrations were 6q deletions (30%), trisomy 18 (15%), 13q deletions (13%), 17p (TP53) deletions (8%), trisomy 4 (8%), and 11q (ATM) deletions (7%) [13]. Deletion of 6q, 11q, and trisomy 4 was associated with poor clinical and biological parameters but was not associated with a decreased survival rate [13]. TP53 deletions have an increased correlation with poor clinical outcomes, as patients with this deletion had a short progression-free survival and short disease-free survival [13]. Copy number changes were identified in nearly 80% of WM cases, notably the inactivation of TNFAIP3 and TRAF3, which are genes involved in the regulation of the NF-kB signaling pathway [12].

More recently, whole-genome sequencing of 30 patients with WM was performed and the MYD88*^L265P^* somatic variant was identified in all patients with positive family history and 86% of sporadic cases [4]. Most notably, the presence of the MYD88^L265P^ somatic variant is rare in patients with MGUS, multiple myeloma, splenic zone marginal lymphoma, and healthy patients, allowing it to serve as a differential factor in diagnosis [4].

## 3. Tumor Microenvironment

The tumor microenvironment has recently become an emerging area of research, with a growing number of studies looking at the tumor microenvironment in not only WM but in other cancers as well. Homing to the bone marrow is a key characteristic of WM and the mechanisms by which WM cells home to the bone marrow have been investigated. Stromal-derived factor-1 (SDF-1), a chemokine responsible for in vitro migration of WM cells, is found highly expressed in the bone marrow of WM patients [14].

The bone marrow is made up of a collection of immune and non-immune cells, including T-cells, B-cells, macrophages, myeloid-derived suppressor cells, mast cells, mesenchymal stem cells, osteoclasts, osteoblasts, natural killer cells, and dendritic cells [15] (Figure 1). While the full function and mechanism of these cells in the progression of WM have not been described, some efforts have been made to quantify the importance of these cells in WM prognosis. Recently, reports on the role of mast cells, T-cells, monocytes, and endothelial cells in WM have been published. Mast cell hyperplasia is a characteristic of WM. It has been previously demonstrated that mast cells in the bone marrow of WM patients induce the proliferation of malignant B cells through CD40L and CD40 interactions [16].

T-cells have also been examined in WM and the expression of PD-1 and the ligands PD-L1 and PD-L2 have been characterized. PD-L1 and PD-L2 gene expression was induced by IL-21, interferon-γ, and IL-6 expression in WM cell lines and patient bone marrow cells. Increased expression of PD-L1 and PD-L2 in the bone marrow of WM patients increased the proliferation of malignant B cells and reduced T-cell proliferation [17].

Bone marrow stromal cells (BMSC) are a heterogenous population that have been shown to play an important role in normal and malignant cell biology [18]. Mesenchymal stem cells (MSCs) serve as the progenitor for most bone marrow stromal cell populations, including osteoblasts, chondrocytes, fibroblasts, endothelial cells, and myocytes [18]. In WM, BMSCs have been shown to regulate the proliferation of tumor cells while contributing to increased drug therapy resistance [19].

Endothelial cells have been shown to increase WM cell adhesion and proliferation through the Ephrin receptor B2 (Eph-B2), which is found upregulated in WM cells [20]. The Eph-B2 receptor was found to be activated in WM patients compared with healthy samples. Endothelial cells in the bone marrow express high levels of Ephrin-B2 ligand. Blocking of either Ephrin-B2 or Eph-B2 inhibited the increased adhesion and proliferation caused by the endothelial-WM cell interaction [20].

## 4. Mechanisms of Disease Progression

### 4.1. Proliferation

IL-21 is a type I cytokine commonly found in the WM tumor microenvironment that rapidly induces the phosphorylation of STAT3 in WM cells (Figure 2) [21]. MWCL-1 cells cultured in the presence of IL-21 for 72 h in vitro demonstrated a dose-dependent increase in both WM cell proliferation and phosphorylated STAT3 levels in those cells [21]. Additionally, in MWCL-1 cells, 10 min of stimulation with IL-21 displayed a significant increase in the phosphorylation of STAT3 [21]. Treatment with a STAT3 inhibitor eliminated the IL-21-mediated increase in proliferation [21].

Fibroblast growth factor receptor 3 (FGFR3) is a member of the FGFR family that interacts with fibroblast growth factor 3 (FGF3), inducing a cascade of downstream signals that influence cell proliferation. This is well documented in many types of cancer, including tongue, colorectal, breast, bladder, and oral cancers [22,23,24,25,26,27]. In WM, the expression of FGFR3 on CD19+ cells from WM patients was greater than the expression on B cells from healthy subjects, and FGFR3 was also overexpressed in the cell lines BCWM.1 and MEC-1 [28].

In cancer, overexpression of the Akt and mTOR pathways play an important role in the progression of malignancies through the phosphatidylinositol-3-kinase (PI3K)/Akt/mammalian target of rapamycin (mTOR) pathway. This pathway can enhance cell survival by inhibiting cell death and stimulating cell proliferation [29,30]. The activation of this pathway ultimately leads to growth, angiogenesis, resistance to apoptosis, and therapy resistance [31,32]. In WM, constitutive activation of the PI3K/Akt pathway exists and leads to increased cell proliferation and resistance to apoptosis [33]. Phosphatases and tensin homolog (PTEN) are haploinsufficient tumor suppressors; therefore, partial loss-of-function mutations can have a dramatic effect on cancer progression. PTEN acts to deactivate the PI3K/Akt/mTOR pathways, therefore loss-of-function can lead to constitutive activation. Studies in mouse models have shown that even a small reduction in PTEN expression can significantly increase cancer risk [34,35]. Unfortunately, PTEN loss-of-function mutations are frequent in human cancers, leading to the perpetual activation of AKT. Furthermore, the role of PTEN in WM has not been reported.

IL-6 plays an important role in normal B cell proliferation and maturation and in B-cell malignancies including diffuse large B-cell lymphoma [36], Hodgkin lymphoma [37], and multiple myeloma [38], where it has been shown to regulate the growth of malignant cells. Previous studies have shown that serum IL-6 levels are increased in patients with WM compared to healthy patients [39]. IL-6 has shown a significant upregulation of IgM secretion by WM cells through the CCL5-IL-6-IgM axis in the TME [40,41]. CCL5 signaling has been shown to induce the expression of the transcription factor GLI2 through the PI3K-AKT-IκB-p65 pathway. GLI2 is required to modulate IL-6 expression in vitro and in vivo through this pathway [40]. Targeting the IL-6 receptor with Tocilizumab to block IL-6 effects on WM tumor cells was shown to reduce IgM levels and deter tumor growth in WM, while not inducing toxicity [42]. This suggests that blocking IL-6 may provide therapeutic efficacy in WM. Despite this, targeting IL-6 in WM patients has not been investigated.

The role of bone marrow stromal cells has been extensively studied in WM and are attributed to the growth of WM cells [33,43,44,45]. Ephrin-B2 was demonstrated to be highly expressed on endothelial cells from the bone marrow of patients with WM compared with healthy controls [20] and activation of the Eph-B2 receptor did not directly increase the proliferation of WM cells, but it increased the adhesion of WM cells to endothelial cells, promoting WM cell proliferation [45]. This increase in WM cell proliferation is dependent on downstream activation of focal adhesion kinase (FAK) and Src and inhibition of ephrin-B2 on endothelial cells or inhibition of Eph-B2 on WM cells reduced the adhesion of WM cells to endothelial cells, preventing the proliferative induction from occurring [45].

B-lymphocyte stimulator (BLyS) is a TNF family member expressed by dendritic cells, neutrophils, monocytes, and macrophages and has been shown to be necessary for normal B-cell development. BLyS binds to the receptors B-cell-activating factor of the TNF family receptor (BAFF-R), transmembrane activator and CAML interactor (TACI), and B-cell maturation antigen (BCMA) in WM patients. Expression of BLyS in WM patient bone marrow and elevated serum BLyS levels have also been noted, as well as upregulated IgM secretion upon BLyS addition. In vitro, BLyS was shown to enhance the proliferation and survival of WM cells [46].

Bone marrow mast cells are commonly associated with malignant cells in patients with WM. CD40 ligand (CD40L/CD154) is an inducer of B-cell proliferation and is expressed on malignant cell-associated mast cells in 94% of WM patients, in contrast with 0% of healthy patient mast cell samples. It was found that the co-culture of mast cells and lymphoplasmacytic cells (LPC) induced LPC proliferation and tumor colony formation [16]. Increased Erk phosphorylation and cell growth in malignant B-cells co-cultured with CD40L-expressing stromal cells have also been reported. GLI2 induced increased CD40L expression and GLI2 knockdown decreased CD40L expression. GLI2 has been shown to directly bind to and regulate the activity of the CD40L promoter [47].

### 4.2. Survival

Myeloid differentiation factor 88 (MYD88) L265P somatic mutation is present in 91% of WM/LPL patients, per whole genome sequencing results [48,49]. The presence of MYD88 L265P has also been reported in IgM MGUS [50], mucosa-associated lymphoid tissue lymphoma (9%) [51], and diffuse large B-cell lymphoma [52]. Inhibiting MYD88/IRAK signaling induced apoptosis of MYD88 L265P-expressing WM cells by blocking MYD88 homodimerization, an essential process for IRAK1 and IRAK4 signaling (Figure 2). This treatment induced significant apoptosis in BCWM.1, MWCL-1 cell lines as well as primary WM patient cells. Induction of apoptosis did not occur without the MYD88 L265P mutation [48]. Due to the activation of NF-kB, increased anti-apoptotic Bcl-xL expression has been observed in both MYD88 L265P and MYD88 L265RPP mutations, promoting increased survival of malignant cells [53].

### 4.3. Angiogenesis

Angiogenesis plays an essential role in wound healing and bone repair and regeneration. This process forms new blood vessels from existing ones, which allow the body to re-establish normal blood flow and oxygen/nutrient/growth factor delivery to the injured or proliferating area [54,55,56,57]. In cancer, tumor cells can develop an angiogenic phenotype through the upregulated pro-angiogenic or downregulated anti-angiogenic pathways [58,59]. This causes endothelial cells to enter a rapid growth phase, forming new blood vessels, and providing nutrients, oxygen, and growth factors to the tumor cells [60]. This process is often rushed in cancer and endothelial cells do not have the time to form perfect blood vessels, leading to leaky, disorganized blood vessels [61,62]. This is an essential step of disease progression and serves to initiate the process of metastasis in many types of cancer [15,61]. VEGF is a well-established growth factor, known for its role in both physiological and pathological angiogenesis. VEGF-A is the main member of the VEGF family and plays a key role in promoting angiogenesis during embryonic development and tissue repair under physiological conditions (Figure 2) [57]. In cancer, VEGF-A production from tumor cells results in an angiogenic switch, leading way to vasculature growth and as a result, tumor growth and metastasis [57]. As the tumor mass increases, the oxygen availability of decreased and hypoxia occurs, leading to the release of proangiogenic factors such as VEGF-A [57]. Angiopoietin-1 (Ang-1) and its antagonist, angiopoietin-2 (Ang-2) serve as the ligands for receptor tyrosine kinase Tie-2 and play a critical role in angiogenesis in both physiological and malignant conditions [63]. Fibroblast growth factors (FGF) are a family of heparin-binding growth factors. Basic FGF (bFGF) interacts with endothelial cell surface receptors and has pro-angiogenic activity [64]. The crosstalk between bFGF, VEGF, and other inflammatory cytokines plays an important role in mediating angiogenesis in the tumor microenvironment.

In WM, the bone marrow microvessel density is only elevated in 30-40% of patients [65]. In a study of 56 patients with WM, it was reported that increased levels of angiogenin, vascular endothelial growth factor (VEGF), vascular endothelial growth factor A (VEGFA), and basic fibroblast growth factor in sera of patients, compared with healthy controls [66]. A lower level of the angiogenesis antagonist, angiopoetin-1 (Ang-1), was also reported in WM sera versus healthy controls [66].

### 4.4. Hypoxia

Hypoxia plays an important role in the progression of many malignancies and activated hypoxia pathways are strongly associated with adverse prognosis in cancer [15]. Tumor hypoxia in multiple myeloma activates HIF1α, which promotes cell survival, motility, invasiveness, drug resistance, and neoangiogenesis [67,68] and is associated with a more aggressive tumor [69]. In multiple myeloma, the egress of bone multiple myeloma cells from the bone marrow into the circulation and into new niches was also demonstrated [70].

In a study demonstrating hypoxia in WM cells, the WM cell line, BCWM.1, was genetically engineered to express luciferase and mCherry fluorescent protein. The cells were injected into SCID mice via the tail vein and allowed to grow for 3 weeks to establish tumor burdens in the bone marrow of the mice [69]. This growth in the bone marrow was confirmed by flow cytometry. The mean fluorescent intensity (MFI) of hypoxia marker pimonidazole hydrochloride signal was analyzed and a direct correlation between the tumor burden in the bone marrow and hypoxia in the WM cells was found. Other cells in the bone marrow were tested for hypoxic signs as well and found that the mCherry-negative population was less hypoxic than the WM cells, but still showed hypoxic signs, and hypoxic signs were more greatly shown at higher tumor burdens [69]. In addition, the effect of tumor hypoxia on the egress of WM cells from the bone marrow was tested and a direct linear correlation between the hypoxia in the bone marrow and the number of circulating WM cells was found [69]. This indicated that the mechanism of WM cell entry into circulation is regulated by hypoxia.

Hypoxia also plays a major role in regulating WM cell proliferation. BCWM.1 and MWCL.1 WM cell lines were exposed to normoxic and hypoxic conditions for 24 h in vitro and found that after 24 h of normoxia, the BCWM.1 and MWCL.1 cells had nearly doubled, and the hypoxic cells only increased by 1.3-fold [69]. This suggests that hypoxic conditions do not promote WM cell growth but play a role in other aspects of WM biology.

### 4.5. Epithelial–Mesenchymal Transition

The epithelial–mesenchymal transition (EMT) describes a process in which epithelial cells lose their epithelial characteristics and gain a mesenchymal phenotype [71]. This process can lead to increased invasiveness of the cancer cells, leading to overall metastasis [72]. This process allows cancer cells to leave the primary tissue site, enter the bloodstream, and infiltrate other tissues [15].

In a study of WM cells and hypoxia, EMT markers E-cadherin, CXCR4, and VLA-4 were assessed via flow cytometry to determine the effect of hypoxia on EMT in WM. BCWM.1 cells were exposed to either normoxic or hypoxic conditions for 24 h, then analyzed for expression of EMT markers by flow cytometry [69].

The adhesion ability of WM cells to bone marrow stromal cells and to each other was assessed in vitro and incubation of BCWM.1 or MWCL.1 cells in hypoxic conditions reduced their adhesion to a bone marrow stromal cell monolayer by 50% and 25%. This decrease in adhesion was linked to reduced expression of the epithelial marker E-cadherin in WM cells [69].

### 4.6. Tumor Spreading and Tissue Infiltration

Ephrin receptors (Eph) represent the largest family of receptor tyrosine kinases (RTK) and are divided into 2 classes: Eph-A and Eph-B, depending on their affinity to ligands ephrin-A and ephrin-B, and they play a critical role in embryogenesis by positioning cells and modulating cell morphology [73,74,75]. As these receptors are not typically found in adult tissue, the presence of EphA1/A2 and ephrin-A1 has been correlated with tumor malignancy and prognosis. The role of these receptors in cancer is still unknown, as they have been found over-expressed in some cancers, but downregulated in others. For example, higher ephrin-A1 expression in liver and colorectal cancer is associated with a worse prognosis [76,77], but in stage I non-small cell lung cancer patients, higher expression levels of EphA2 and ephrin-A1 improved their prognosis [78]. In WM patient samples, the Eph-B2 receptor was found to be overexpressed in WM cells. Inhibition of ephrin-B2 on endothelial cells led to decreased adhesion of WM cells to endothelial cells, therefore decreasing proliferation, cell-cycle progression, and tumor progression in WM cells [20]. In a study looking at the effect of Eph-B2 in WM cells, it was found that inhibition of Eph-B2 on WM cells reduced bone marrow infiltration by WM cells [45].

### 4.7. Disease Progression Complications

Bing–Neel syndrome (BNS) is a rare complication of WM. Two types of BNS exist, diffuse and tumoral form. In diffuse form, malignant cells are found in the leptomeningeal space, periventricular white matter, or the spinal cord. The tumoral form is characterized by an intraparenchymal mass or nodular lesion [79]. BNS is rare, with only 1% of patients showing BNS during the disease progression. The treatment of BNS requires drugs with successful infiltration into the central nervous system, such as fludarabine, methotrexate, and cytarabine. Ibrutinib has shown some CNS-penetrating properties and may have a therapeutic role in treating BNS [80].

## 5. Proposed Therapies

There is no standard therapy for the treatment of WM [28] and only two FDA approved treatments, Ibrutinib and Zanubrutinib, exist [81]. Most treatments were originally derived from other lymphoproliferative diseases such as multiple myeloma and chronic lymphocytic leukemia [82].

Due to the crucial role of B cell receptor (BCR) signaling in B cell development and pathogenesis of B cell malignancies, efforts to drug the BCR signaling pathway have been extensively researched for the treatment of B cell malignancies [83]. Due to the criticality of Bruton’s tyrosine kinase (BTK) in BCR signaling, BTK is an important therapeutic target and as a result, several BTK inhibitors have been developed and have shown remarkable results in treating other B cell malignancies, such as chronic lymphocytic leukemia (CLL) [84,85], mantle cell lymphoma (MCL) [84], marginal zone lymphoma (MZL) [86], and Waldenström macroglobulinemia (WM) [86].

### 5.1. BTK Therapy

Due to the abnormal B-cell receptor signaling in disease progression in WM, Bruton’s Tyrosine Kinase inhibitors have proved successful in treating these malignancies [87]. BTK inhibitors work by blocking BTK activation, therefore inhibiting NF-κB and MAP kinase activation, leading to reduced survivability and proliferation. In a 2015 study of Ibrutinib in 63 symptomatic patients with WM who had at least one previous treatment was conducted. Ibrutinib was administered orally (420 mg daily) until disease progression or unacceptable toxic effects were observed. After Ibrutinib treatment, median serum IgM levels decreased (3520 mg/dL to 880 mg/dL), median hemoglobin levels increased (10.5 g/dL to 13.8 mg/dL), and bone marrow involvement decreased (60% to 25%). The overall response rate (ORR) was 90.5% and the major response rate (MRR) was 73.0%. These rates were dependent on the mutational status of the patients, with the highest rates among patients with MYD88 L265PCXCR4^WT^ (100% overall response rate and 91.2% major response rate). Patients with MYD88 L265PCXCR4^WHIM^ experienced an ORR of 85.7% and MRR of 61.9% and patients with MYD88 WTCXCR4^WT^ status had an ORR of 71.4% and MRR of 28.6% [3,88].

Zanubrutinib, a selective BTK inhibitor, was FDA approved in 2021 for the treatment of WM. In a phase 1/2 study of patients with WM, either treatment naïve or relapsed/refractory, the overall response rate was 95.9% at 36 months post-treatment initiation [89]. In a randomized phase 3 trial of Zanubrutinib versus Ibrutinib in WM, patients with MYD88 L265P disease were randomly assigned to treatment with Ibrutinib or Zanubrutinib. More patients in the Zanubrutinib group (28%) versus the Ibrutinib group (19%) achieved a very good partial response, and side effects of BTKi therapy, including contusion, diarrhea, edema, atrial fibrillation, and other adverse effects leading to treatment discontinuation were lower in Zanubrutinib patients versus Ibrutinib [90].

Noting the mutational status of WM patients has proved to be critical in BTKi treatment. In patients with MYD88 L265P mutation, favorable prognostic effects were noted. In a study of 219 patients, patients with the MYD88 L265P mutation status led to favorable overall survival in patients who received BTKi treatment. TP53 mutation was associated with significantly poorer overall survival and progression-free survival in treated patients. The CXCR4NS/MS mutation was associated with a significantly shorter time to treatment and 93.3% of patients had an intermediate/high-risk International Prognostic Scoring System for WM score [91].

As mutational status research continues to guide our clinical decision-making, it is becoming easier for physicians to prescribe treatment regimens that will be most beneficial for patients.

### 5.2. Anti-CD20 Therapy

Rituximab is a mouse anti-human CD20 monoclonal antibody used to treat low-grade and follicular lymphoma, CLL, diffuse large B-cell lymphoma and WM [92]. Anti-CD20 therapy has shown effective due to the high presence of the CD20 antigen on malignant B-cells, allowing for effective targeting and depletion of CD20+ malignant B-cells. The most common clinical combination of rituximab and other drugs involves alkylating agents such as cyclophosphamide or bendamustine, or the nucleoside analog, cladribine in WM. Novel proteasome inhibitors, thalidomide, and everolimus have shown positive therapeutic potential in WM, but mostly in combination with rituximab. Rituximab/chemotherapy is one of the cornerstones in the treatment of relapsed patients with WM [4].

### 5.3. Combination Therapy

A study was completed in 2017 on the effects of combination treatment of Ibrutinib-rituximab and the results were promising. One hundred fifty patients were randomly assigned to receive either Ibrutinib-rituximab or placebo-rituximab. At 30 months post-treatment, the progression-free survival rate for the Ibrutinib-rituximab group was 82%, versus 28% with placebo-rituximab. These results were also independent of the MYD88 or CXCR4 genotypes. The rate of major response was also significant, with 73% in the Ibrutinib-rituximab group experiencing this, versus only 41% in the placebo-rituximab group [10]. Additionally, the median IgM level declined more rapidly and significantly; after only 4 weeks of treatment, the median IgM level was reduced from the baseline by 56% in the Ibrutinib-rituximab group, compared to an increase of 6% in the placebo-rituximab group. In a five-year follow-up of this study, the Ibrutinib–rituximab combination remained the superior treatment option over placebo-rituximab in both treatment-naïve and previously treated patients, regardless of genomic factors [93].

In a randomized trial with 64 patients, R-CHOP (rituximab, cyclophosphamide, doxorubicin, vincristine, prednisone) was a well-tolerated and effective first-line treatment for WM [94]. In this study, R-CHOP was compared to CHOP treatment and R-CHOP patients had a significantly higher overall response rate (94% vs. 67%) in the LPL patients and in the WM subgroup (91% vs. 60%). R-CHOP also induced a significantly longer time to treatment failure than CHOP (63 months vs. 22 months) [94]. This study was also conducted by the Eastern Cooperative Oncology Group trial and showed that 91% of R-CHOP patients achieved a median partial response (PR) of 1.6 months, with a median follow-up time of 18.3 months. These studies indicated that rituximab with CHOP is highly effective and well-tolerable in some patients. Medically fit patients can generally tolerate R-CHOP well, but in many patients, R-CHOP is too toxic because of the myelosuppressive effects [4].

In attempts to minimize the toxicity of R-CHOP treatment, different treatment combinations were assessed. The outcomes of symptomatic patients with WM who received R-CHOP with CVP-R (cyclophosphamide/vincristine/prednisone/rituximab), or CP-R (cyclophosphamide/prednisone/rituximab) were studied. The baseline characteristics were the same for all three cohorts; age, previous therapies, bone marrow involvement, hematocrit, platelet count, and serum B2-microglobulin were all similar. Serum IgM levels were higher in patients treated with R-CHOP, however. The overall response rates and complete response rates were as follows: R-CHOP (96% and 17%), CVP-R (88% and 12%), and CP-R (95% and 0%). CP-R was the safest treatment, with adverse reactions the least common [95].

In a study of Dexamethasone followed by rituximab intravenously on day 1 and cyclophosphamide on days 1–5 (DRC), this was found to be highly effective in a phase II trial in 72 previously untreated patients with symptomatic WM. The overall response rate was 96%, and the major response rate was 87%. DRC is a promising chemotherapeutic regimen, due to the low comparative cost of DRC treatment, versus other therapeutic combinations [96,97].

The combination of subcutaneous cladribine with rituximab was investigated in patients who were either treated or untreated previously. Rituximab was administered on day 1, followed by subcutaneous cladribine for 5 consecutive days, administered monthly for four cycles. The median follow-up was 43 months, and the overall response rate was 89.6%, with 7 complete responses, 16 partial responses, and 3 minor responses [98].

Bendamustine plus rituximab (BR) therapy has become increasingly popular as a treatment for WM and other B-cell lymphomas. In a study of BR on WM, after a median follow-up of 23 months, disease progression was present in six patients, but overall survival and progression-free survival were 97.1% and 87%, respectively. The presence of MYD88 and CXCR4 mutations did not impact survival in the BR treatment group, indicating positive results for this treatment option [99].

### 5.4. Novel Therapies

The management of WM has advanced tremendously with recent genomic findings that can help guide treatment approaches [93]. The current diagnosis of WM requires bone marrow infiltration by lymphoplasmacytic lymphoma cells, increased IgM levels, and the noted presence or absence of the MYD88 L265P mutation [100]. Bone marrow involvement and serum levels of IgM, albumin, and β2-microglobulin are often used to guide proper treatment initiation time [100].

Novel covalent and noncovalent BTK inhibitors, BCL2 antagonists, and CXCR4-targeting agents are being developed and show promising futures for the treatment of WM. BTK inhibitors are Tirabrutinib, Vecabrutinib, LOXO-305, and ARQ-531. The BCL2 antagonist is Venetoclax, and the CXCR4-targeting agents are Ulocuplumab and Mavorixafor.

Tirabrutinib is a second-generation Bruton’s tyrosine kinase inhibitor with greater selectivity than Ibrutinib. In a phase II study of Tirabrutinib on patients with treatment-naïve or relapsed/refractory WM, it was shown that Tirabrutinib monotherapy is a highly effective therapy option for both untreated and relapsed/refractory WM patients [101].

Vecabrutinib is a selective, reversible, and non-covalent BTK inhibitor that has been studied in multiple B-cell malignancies [102]. A phase 1B study, conducted in 2019 on twenty-one patients with CLL, two patients with MCL, three patients with WM, and one patient with MZL showed promising results. No patients experienced serious adverse events from Vecabrutinib treatment, and all patients showed improvement in symptoms and decreased tumor burden [102].

#### 5.4.1. PI3K Inhibitors

Due to the commonality of WM patients with mutations in MYD88 or CXCR4, which activate downstream PI3K signaling [103], the PI3K/Akt signaling pathway is a promising target for WM treatment. Idelalisib is a potent inhibitor of the phosphatidylinositol 3-kinase isoform-delta (PI3Kδ). Idelalisib is a preferred treatment because of related bleeding episodes in patients with B-lymphoproliferative disorders receiving Ibrutinib [104]. However, Idelalisib has been cited to play a role in significant adverse effects, including hepatic and infectious effects [105,106]. Buparlisib (BKM120), a 2,6-dimorpholino pyrimidine derivative, is a potent pan-class I PI3K inhibitor with 50-fold selectivity against other protein kinases. Treating WM cells with buparlisib decreased the activation of signaling proteins pAkt, pS6R, pP70S6K, and p-mTOR in a dose-dependent manner. Increasing concentrations of buparlisib also induced apoptosis and significantly decreased the rate of tumor progression [107]. A siRNA knockdown of the different isoforms of PI3K represents the importance of PI3K inhibitors, where the survival rate of treated WM cells was 50% of that of the control [108].

#### 5.4.2. AKT Inhibitors

Perifosine is a novel Akt inhibitor belonging to a class of lipid-related alkylphospholipids [108,109]. Perifosine inhibits proliferation and induction of apoptosis in WM cells in vitro [110] and produces a significant reduction in WM tumor growth, demonstrated in a subcutaneous murine xenograft model, through its inhibition of Akt phosphorylation and downstream targets [110]. In vitro migration and adhesion and in vivo homing of WM cells to the BM microenvironment are also reduced with Perifosine treatment [110]. One of the first phase II clinical trials studying Perifosine activity found that all patients exhibited response, and the median time to progression and progression-free survival was 12.6 months, longer than most other targeted agents in similar conditions. Perifosine also reduced the expression of several genes involved in adhesion and migration, as well as molecules important in the NF-κB pathway. The most common side effects included gastrointestinal symptoms, fatigue, cytopenias, and flare of arthritis/joint effusions [47].

Growth receptor stimulation in the presence of Perifosine may lead to Phospholipase C (PLC) and RTK activation [110], which induces PIP2 activation, leading to activation of PKC and inducing growth stimulation. PKC is associated with the activation of downstream RAF proto-oncogene serine/threonine-protein kinase (c-Raf) and Mitogen-activated protein kinase (MAPK)/ERK. Inhibition of ERK or MAPK by U0126 in combination with Perifosine leads to synergistic inhibition of growth and induction of cell death.

Bortezomib, an inhibitor of the ERK/MAPK pathway, and Perifosine, an Akt inhibitor that induces activation of the ERK/MAPK pathway, were tested together in vitro. The combination was able to inhibit the pathway activation of the other agent and led to a decrease in both p-Akt and p-ERK activity. Perifosine and bortezomib significantly inhibit the phosphorylation of downstream target proteins of Akt, phospho-S6 ribosomal protein, and phospho-GSK-3α/β, in a dose-dependent manner. The combination of Perifosine and bortezomib increased cytotoxicity from 31% for Perifosine alone to 59–69% in combination. This significantly decreases the survival of WM cells [46].

In a phase II trial of Perifosine, the median progression-free survival was 12.6 months and 54% of patients maintained stable disease. No patients achieved complete response, but 35% of patients did exhibit at least a minimal response. In vivo inhibition of pGSK activity was noted, warranting further studies of perifosine in combination with other therapeutics [47].

NVP-BEZ235 inhibits phosphorylation of Akt, GSK3α/β, and ribosomal protein S6 in a dose-dependent manner. It also inhibits phosphorylation of mTOR, and its downstream targets, p70S6 and 4EBP1. Apoptosis is induced through intrinsic and extrinsic apoptotic pathways through the cleavage of PARP, caspase-3, caspase-8, and caspase-9. NVP-BEZ235 through targeting the PI3K/Akt signaling pathway exerts anti-tumor activity. NVP-BEZ235 also abrogates bone marrow-derived mesenchymal stem/stromal cell (BMSC) adhesion-induced phosphorylation of Akt and mTOR. This indicates that NVP-BEZ235 can exert its antitumor activity even when WM cells were in contact with the BM mitigating the pro-tumorigenic adhesion effects. Specifically, NVP-BEZ235 inhibits phosphorylation of focal adhesion kinase, paxillin, and cofilin, proteins that act as key regulators of adhesion and cell migration. Treatment of WM cells with NVP-BEZ235 resulted in significant inhibition of homing of WM cells to the BM at different time points after injection [35].

#### 5.4.3. mTOR Inhibitors

Rapamycin and its analogs function as allosteric inhibitors of TORC1 and cause up to 45% partial remission of WM in patients [111]. Abrogation of feedback circuits by rapamycin results in increased phosphorylation of Akt, leading to enhanced survival and chemoresistance [112,113,114,115,116]. However, these drugs only partially inhibit TORC2 in most tumor cells, and often leave the signaling cascades downstream of both TORC1/2 complexes active [117,118,119]. Drug-related pneumonitis is also a well-recognized side-effect of mTOR inhibitors [120].

Everolimus is a rapalog that inhibits mTOR and exerts antitumor activity [111]. An in vitro study found that Everolimus inhibited cell proliferation, regardless of the presence of endothelial cells, and induced cytotoxicity in a dose-dependent manner. Everolimus also inhibited WM cell growth in the BM milieu, migration toward SDF-1, and cell adhesion. Everolimus-induced reduction of endothelial cell morphogenesis caused upregulation of pro-survival signaling pathways in WM cells, while endothelial cells increase the proliferative rate of WM cells by 32%. At higher concentrations of the drug, tube formation was blocked almost completely [121]. The overall response rate (ORR) in a phase II study of Everolimus was high with initial therapy, but the ORR was between 30% to 40% in the salvage setting. The median progression-free survival (PFS) was 21 months in a separate study, where patients with major responses exhibited longer PFS than those with minor responses or stable disease [122]. In relapsed WM, Everolimus results compare favorably with other single-agent trials [123]. Everolimus was found to have common but manageable [111,123] toxicities. Grade 3 or higher toxicities, primarily hematologic toxicity, have been observed in 67% of patients. Symptoms improved with the addition of steroids and reduction in Everolimus dose [123]. A prospective, multicenter study of WM patients found that rapid increases in serum IgM levels were common following discontinuation of Everolimus [123]. A separate study noted that 58% of patients developed relapsed or refractory WM31.

Everolimus does not completely inhibit TORC2 in most cells, however. This leads to enhanced survival and chemoresistance through loss of S6K inhibition and Akt activation. The lack of complete response from Everolimus treatment suggests that some of the lymphoplasmacytic cells might be resistant to mTOR inhibition that combination therapies may target. When bortezomib was used in combination with Everolimus, the cytotoxicity of WM cells almost doubled from 24%. Significant inhibition of NF-κB/p65 DNA-binding activity (84%) was documented in WM cells exposed to the combination of Everolimus and bortezomib compared to Everolimus alone [121]. An increase in specific lysis was also observed in combination treatment of Everolimus, rituximab, and bortezomib, which increased to 73.1%. Significant inhibition of phosphorylation of S6R, an mTOR downstream targeted protein, was also observed [121].

TAK-228 (formerly MLN0128/INK128) is a selective adenosine triphosphate (ATP) site kinase inhibitor of both TORC1 and TORC2 [124,125]. A study found that two WM patients treated with TAK-228 achieved stable disease (SD), and one achieved a partial response. However, the study was limited to only 4 WM patients [126].

#### 5.4.4. PKC Inhibitors

Enzastaurin, an acyclic bisindolylmaleimide derived from Staurosporine, is a PKCβ-selective inhibitor that inhibits the PI3K/Akt pathway acting as an Akt competitor [127]. Akt phosphorylation is inhibited in a time-dependent fashion by Enzastaurin and subsequently induces ERK phosphorylation. This also inhibits VEGF–induced angiogenesis [128]. Bone marrow cytotoxicity has also been documented with Enzastaurin treatment. In a xenograft mouse model of human WM, Enzastaurin significantly inhibited Akt and GSK3β phosphorylation in tumor cells and inhibited the growth of tumor WM cells in a SCID subcutaneous tumor model [129]. Enzastaurin enhanced rituximab, bortezomib, fludarabine, and dexamethasone antitumor activity, suggesting that combining these agents may be therapeutically useful [129].

#### 5.4.5. SYK Inhibitors

SYK inhibitors include Tamatinib (R406), the active metabolite of fostamatinib [130], and entospletinib. Studies in vitro showed that Tamatinib and entospletinib blocked SYK, STAT3, and Akt phosphorylation in MYD88 L65P WM cells [131]. Fostamatinib reduced the viability of WM cell lines by inducing cell cycle arrest and apoptosis, and dramatically reduced phospho-Ser473 Akt levels by up to 95–98% [132].

#### 5.4.6. IRAK Inhibitors

R191, a newly developed IRAK1/4 inhibitor, has been studied in pre-clinical WM models. R191 exposure decreased Akt activation and downstream Glycogen synthase kinase 3α/β (GSK3α/β) activation. Other Akt signaling intermediates showed decreased activation, including PDK1, mTOR, and S6, consistent with an inhibitory effect on the entire pathway. Cells expressing a constitutively active Akt mutant were more resistant to R191, while cells expressing dominant negative Akt showed slightly increased sensitivity [133]. A combination of R191 and Akt inhibitor Afuresertib was able to enhance the reduction of viability in cell lines [134].

#### 5.4.7. SHIP1 Inhibitors

The use of miRNA155 knockdown has shown promising results in vitro, as the miRNA155 normally targets SHIP1 and acts as a negative regulator of the PI3K/Akt and mTOR pathway [135]. miRNA155 knockdown strongly inhibits ERK, Akt, GSK3α/β, and S6R phosphorylation [136]. miRNA-155 demonstrated a link between NF-κB and PI3K/Akt signaling cascades [137,138]. This complements evidence that following TNFα-induced activation of PI3K, Akt activation is responsible for phosphorylation of IKKα and subsequent activation of NF-κB.

#### 5.4.8. HMG-CoA Inhibitors

Statins, a family of 3-hydroxy-3-methyl-glutaryl-CoA reductase inhibitors, act by interfering with the mevalonate pathway [139]. Simvastatin inhibits proliferation and induces cytotoxicity in WM cells, specifically by inhibiting the phosphorylation of Akt and ribosomal protein S6 in a dose-dependent fashion [140].

### 5.5. microRNAs

MicroRNAs (miRNAs) are non-coding RNAs that can influence protein expression through specific targeting of mRNA molecules through base-pairing between the miRNA and the 3′ untranslated regions of the target mRNA. This leads to degradation of the mRNA, or translational repression, leading to decreased protein expression [141].

The role of miRNAs in WM has been studied. miR-23b-3p was found to be downregulated in patients with WM and has been found to target SP1 3′UTR, which positively affects the NF-kB pathway [142] Upon transfection of miR-23b-3p, a decrease in WM cell proliferation and survival was noted. The 13q14 chromosomal region deletion has been noted in 10% of WM patients. This genomic region is host to miR-15a-5p and miR-16-5p, two miRNAs that have been characterized as pro-apoptotic miRNAs by downregulating BCL2. These two miRNAs also have been shown to downregulate the NF-kB pathway, leading to less WM cell proliferation and survival [143].

miR-155-5p is another important miRNA in WM. This miRNA was found upregulated in WM cells, compared to healthy controls. In studies where miR-155-5p was knocked down, proliferation, adhesion, and migration of WM cells were increased, through PI3K/AKT and NF-kB pathways [144].

In experiments where miR-155-5p was knocked down in WM cells and the cells were treated with Everolimus, inhibition of cytotoxicity was observed, indicating the importance of miR-155-5p in not only disease progression but also therapeutic response as well [14].

miRNAs have also been reported as potential biomarkers for WM as well [121,144]. miR-206-3p and miR-9-3p expression levels are upregulated and downregulated, respectively, in WM. A study investigating histone deacetylases (HDACs) and histone acetyltransferases (HATs) in WM cells after downregulation of miR-206-3p and upregulation of miR-9-3p showed that miR-206-3p was found to downregulate KAT6A and miR-9-3p downregulated HDAC4 and HDAC5 [145]. Epigenetic regulation of HDACs and HATs by miRNAs is important, as deregulation of HDACs and HATs in many malignancies is common.

## 6. Summary

Given the complex heterogeneity of Waldenström macroglobulinemia across different patient populations, it is important to consider all possible immunotherapy pathways for successful treatment of WM. Genetic mutations and previous treatment can impact treatment outcomes and various combination therapies may be required for best response. Intra-tumor heterogeneity (ITH) is commonplace in many cancers and has been noted in WM. This phenomenon increases the difficulty of detection and proper diagnosis, as well as complications of treatment. Due to ITH, increased tumor progression, preservation of oncogenic potential, drug resistance, and increased probability of relapse are all important factors to consider when treating WM. The insurgence of tumor sequencing has increased clinical knowledge and led to better outcomes in drug selection and treatment, but single biopsies from one area of the tumor cannot fully capture the heterogeneity of the entire tumor. This may lead to increased drug resistance and maintenance of oncogenic potential. Single-cell sequencing has become more popular in recent years, allowing for the characterization of the complete tumor landscape [146]. For proper treatment, identifying, tracking, and treating the order of mutations may be the future of WM treatment.

While targeted therapy is generally well tolerated, adverse reactions may develop with long treatment periods, making patient-personalized therapy critical for the successful management of WM. Currently, Ibrutinib and Zanubrutinib are the only FDA-approved treatment for WM, but other immunotherapies are creeping into standard practice.

## Figures and Tables

**Figure 1 ijms-23-11145-f001:**
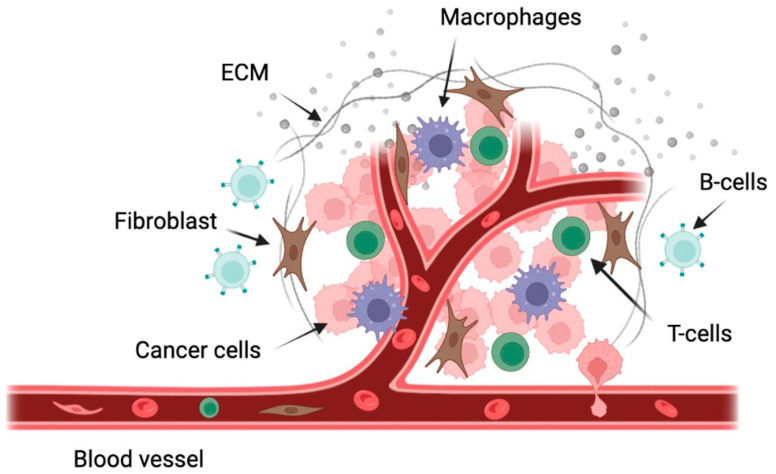
Schematic representation of some of the cells present in the WM tumor microenvironment. This figure was generated using biorender.com.

**Figure 2 ijms-23-11145-f002:**
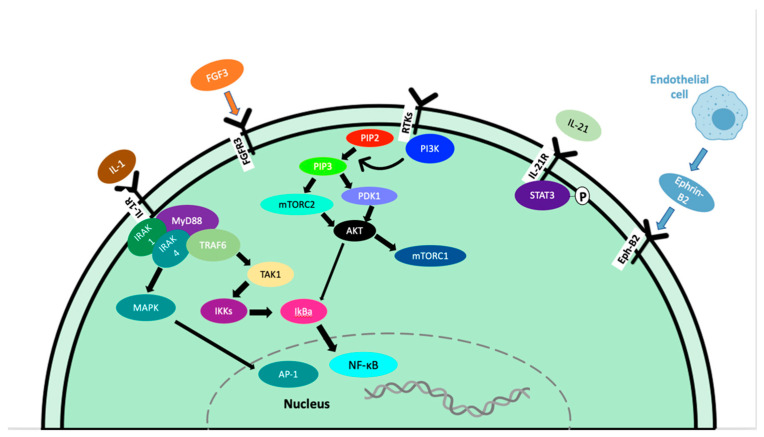
Signaling pathways contributing to tumor progression in WM.

## Data Availability

Not applicable.

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
