# Peer review of "Waldenström Macroglobulinemia: Mechanisms of Disease Progression and Current Therapies"

_ijms, 2022, doi:10.3390/ijms231911145_

Round 1

Reviewer 1 Report

In this Review article, Boutilier et al. aimed at summarizing the current knowledge regarding the progression and therapy strategies of Waldenström Macroglobulinemia (WM). This is a well-written and comprehensive Review, giving a great overview of the available literature. I only have some minor concerns before this manuscript is considered for acceptance:

·       The information regarding the cell surface markers is rather little and should be merged with the Introduction section.

·       A paragraph addressing the existing limitations in all disease aspects, including those of the existing therapies and the necessity for the development of new ones is missing.

·       Future perspectives on WM research should be added as well.

·       Since the authors refer to the molecular background of WM, as well as to new therapies targeting specific components of key pathways for WM progression, it would be interesting to include a paragraph about the implication of microRNAs (miRNAs) in development and progression of WM. These small non-coding RNAs are involved in WM by targeting key molecules for this disease. More information about this topic is summarized in a recent Review by Artemaki et al. (Biomedicines 2021, 9, 333. https://doi.org/10.3390/biomedicines9040333)  

Author Response

Reviewer 1

In this Review article, Boutilier et al. aimed at summarizing the current knowledge regarding the progression and therapy strategies of Waldenström Macroglobulinemia (WM). This is a well-written and comprehensive Review, giving a great overview of the available literature. I only have some minor concerns before this manuscript is considered for acceptance:

  1. The information regarding the cell surface markers is rather little and should be merged with the Introduction section.

We thank the reviewer for the comment. The cell surface markers section has been merged with the introduction.

  1. A paragraph addressing the existing limitations in all disease aspects, including those of the existing therapies and the necessity for the development of new ones is missing.

We thank the reviewer for the comment. Limitations of existing therapies has been added where applicable, and limitations have been discussed in the summary section.

  1. Future perspectives on WM research should be added as well.

We thank the reviewer for the comment. Information about future perspectives has been added to the summary section.

  1. Since the authors refer to the molecular background of WM, as well as to new therapies targeting specific components of key pathways for WM progression, it would be interesting to include a paragraph about the implication of microRNAs (miRNAs) in development and progression of WM. These small non-coding RNAs are involved in WM by targeting key molecules for this disease. More information about this topic is summarized in a recent Review by Artemaki et al.(Biomedicines2021, 9, 333. https://doi.org/10.3390/biomedicines9040333).

We thank the reviewer for the comment. Information about miRNAs can be found in section 5.5, lines 662-732.

Reviewer 2 Report

The scope of the authors is to summarize biologic known mechanism of progression in WM and target therapeutic approach.

Despite the interest in a review on mechanisms of progression in WM, overall i think that the manuscript is not well planned and structured. 

Several mechanism of progression are listed, and then several therapeutic options are listed too.

In this kind of work i would expect that therapeutic approaches would be contextualized with pathogenetic mechanisms. In this way the reader can really understand which mechanisms are important on WM pathogenesis and how to contrast them.

Furthermore, some biologic features that are crucial in WM pathogenesis are only briefly mentioned as well as treatment. I'm talking for example of MYD88L265P and CXCCR4 mutational status and the importance of these genomic profile in treatment with BTKi. Zanubrutinib, that showed activity in CXCR4 whim pts is even not mentioned, as well as pts with WT MYD88 that may benefit from the addition of rituximab to ibrutinib. In the context of CXCR4 mutated pts, treatment with CXCR4 targeting agents, their rationale in addition to BTKi (a trial is ongoing on it) would be of greater interest and should be expanded.

Chemoimmunotherapy section seems not updated. Fludarabine is not used anymore in WM. Similarly, there is only a limited space for cladribine or CHOP, while some regimens as BR or CDR, that are widely used and independent from MYD88/CXCR4 mutational status, are only mentioned.

Minor points:

Introduction:

line 33: do authors mean transformation to aggressive lymphoma?

line 34-35: is a repetition from lines 24-25

Paragraph 2:

line 47: immunocytoma is an obsolete definition

Section 6, line 301: SLL is the same of CLL, and why authors mention colorectal cancer?

Bortezomib is not mentioned

Overall, i think that this work, that could be potentially extremely interesting, should be extensively re-structured and contextualized in order to be useful for the readers.

Author Response

Reviewer 2:

The scope of the authors is to summarize biologic known mechanism of progression in WM and target therapeutic approach.

  1. Despite the interest in a review on mechanisms of progression in WM, overall i think that the manuscript is not well planned and structured.

We thank the reviewer for the comment. The manuscript has been slightly restructured to increase readability.

  1. Several mechanism of progression are listed, and then several therapeutic options are listed too.
  2. In this kind of work i would expect that therapeutic approaches would be contextualized with pathogenetic mechanisms. In this way the reader can really understand which mechanisms are important on WM pathogenesis and how to contrast them.

We thank the reviewer for the comment. The manuscript has been altered to include pathogenic mechanisms where applicable.

  1. Furthermore, some biologic features that are crucial in WM pathogenesis are only briefly mentioned as well as treatment. I'm talking for example of MYD88L265P and CXCCR4 mutational status and the importance of these genomic profile in treatment with BTKi. Zanubrutinib, that showed activity in CXCR4 whim pts is even not mentioned, as well as pts with WT MYD88 that may benefit from the addition of rituximab to ibrutinib. In the context of CXCR4 mutated pts, treatment with CXCR4 targeting agents, their rationale in addition to BTKi (a trial is ongoing on it) would be of greater interest and should be expanded.

We thank the reviewer for the comment. A section on the importance of mutational status in BTKi treatment has been added under section 5.1, after a paragraph about the importance and rationale of Zanubrutinib in BTK therapy. This information can be found on lines 334-353.

  1. Chemoimmunotherapy section seems not updated. Fludarabine is not used anymore in WM. Similarly, there is only a limited space for cladribine or CHOP, while some regimens as BR or CDR, that are widely used and independent from MYD88/CXCR4 mutational status, are only mentioned.

We thank the reviewer for the comment. The mention of fludarabine on line 338 was removed. Additionally, information about BR and CDR was added to lines 393-404.

Minor points:

Introduction:

  1. line 33: do authors mean transformation to aggressive lymphoma?

We thank the reviewer for the comment. The section has been updated and the language has been made clearer to specify that it is aggressive lymphoma of non-Hodgkin type. Lines 32-33.

  1. line 34-35: is a repetition from lines 24-25.

We thank the reviewer for the comment. The repetition was removed.

Paragraph 2:

  1. line 47: immunocytoma is an obsolete definition.

We thank the reviewer for the comment. The term was updated to lymphoplasmocytoid lymphoma.

  1. Section 6, line 301: SLL is the same of CLL, and why authors mention colorectal cancer?.

We thank the reviewer for the comment. The repetition of SLL was removed and colorectal cancer reference was also removed.

  1. Bortezomib is not mentioned.

We thank the reviewer for the comment. Information about Bortezomib can be found between lines 436-444, lines 494-500, and lines 515-517.

  1. Overall, i think that this work, that could be potentially extremely interesting, should be extensively re-structured and contextualized in order to be useful for the readers.

We thank the reviewer for their helpful comments that have significantly enhanced the quality of the review.

Reviewer 3 Report

please see file enclosed

Author Response

Reviewer 3:

The authors report on mechanisms of disease progression and treatment of Waldenström Macroglobulinemia.

The manuscript is mainly well written. However, I have several comments:

  1. The authors state that ibrutinib is the only FDA-approved agent for Waldenström Macroglobulinemia. This is not correct. Zanabrutinib is also FDA-approved for the treatment of this disease since 2021. Why do the author not comment on this important agent despite the fact that they discuss many new agents not introduced into the clinic yet?

We thank the reviewer for the comment. The statement about Ibrutinib being the only approved treatment has been altered to include Zanubrutnib (line 742). Information about Zanubrutnib has been added to section 5.1: BTK Therapy, lines 334-350.

  1. A short section on special types of Waldenström Macroglobulinemia including their treatment options might be helpful (e.g., Bing Neel syndrome).

We thank the reviewer for the comment. Some information about BNS and treatment options has been added to section 4.7.

  1. The authors write that Waldenström Macroglobulinemia is incurable. What is about allo-HSCT?

We thank the reviewer for the comment. Information about allo-HSCT has been added to the introduction after the incurable statement. Due to the high relapse rate, WM is still considered incurable.

  1. Page 2, first paragraph: the authors write that myeloma is characterized by `translocation´. Do the authors refer here to specific IGH-translocations such as t(4;14)?

We thank the reviewer for the comment. This section has been clarified to define IGH-translocation and their importance in IgM myeloma characterization. Specific translocations have been described.

  1. Page 2, middle paragraph: the authors write that Waldenström Macroglobulinemia might be differentiated from multiple myeloma and other lymphoma types by the presence of MYD88 variants. I recommend focusing here more clearly on IgM multiple myeloma (which is very rare) and IgM MGUS in this section.

We thank the reviewer for the comment. We rewrote some of this section and added information about IgM MM and MGUS in this section.

  1. Page 3, last paragraph: the expression `metastasis´ is rarely used in the context of lymphoma/myeloma. Please consider to substitute this expression. The authors write `...expression in live and colorectal..´´ I assume this should be `liver´.

We thank the reviewer for the comment. The misspelling of liver has been corrected. Additionally, the term metastasis has been adjusted to describe the process more accurately in WM and other lymphomas. The term has been updated to “tumor spreading and tissue infiltration”.

  1. Page 7, last paragraph. The authors write that the overall response rate was both 100% in patients with vs. without CXCR4 mutations. However, the response rate was in some groups below 100% if separating by MYD88 and CXCR4 mutational status. This does not sound reasonable. The mutational status (e.g. wt) should also be described more clearly.

We thank the reviewer for the comment. That sentence has been removed to make the section more clear.

  1. Page 11, upper paragraph: there are some numbers behind some statements without brackets. Are these references?

We thank the reviewer for the comment. These numbers have been either turned into references, or removed, depending on the context.

  1. The authors write that there are some in vitro studies assuming that statins might be effective to treat Waldenström Macroglobulinemia. Do the authors have any clinical data in this regard?

We thank the reviewer for the comment. Clinical data regarding some of the therapies has been added where applicable (see lines 548-552).

  1. There are minor linguistic improvements required. Some substances are written either with upper-or lowercase letters (e.g. Ibrutinib vs. ibrutinib). Please unify spelling.

We thank the reviewer for the comment. The spelling of these substances has been unified.

Round 2

Reviewer 2 Report

Line 439-444, are describing the same regimen of lines 423-426

Author Response

We thank the reviewer for the comments. We have addressed them in this revised version of the manuscript.
